# Gancaonin N from *Glycyrrhiza uralensis* Attenuates the Inflammatory Response by Downregulating the NF-κB/MAPK Pathway on an Acute Pneumonia In Vitro Model

**DOI:** 10.3390/pharmaceutics13071028

**Published:** 2021-07-06

**Authors:** Hyun Min Ko, Seung-Hyeon Lee, Wona Jee, Ji Hoon Jung, Kwan-Il Kim, Hee-Jae Jung, Hyeung-Jin Jang

**Affiliations:** 1College of Korean Medicine, Kyung Hee University, 26, Kyungheedae-ro, Dongdaemun-gu, Seoul 02447, Korea; rhgusals93@naver.com (H.M.K.); skyking27@naver.com (S.-H.L.); 97wona@naver.com (W.J.); johnsperfume@gmail.com (J.H.J.); 2Department of Science in Korean Medicine, Graduate School, Kyung Hee University, Seoul 02447, Korea; 3Division of Allergy, Immune and Respiratory System, Department of Internal Medicine, College of Korean Medicine, Kyung Hee University, 23 Kyungheedaero, Dongdaemun-gu, Seoul 02447, Korea; myhappy78@naver.com; 4Department of Clinical Korean Medicine, Graduate School, Kyung Hee University, Seoul 02247, Korea

**Keywords:** Gancaonin N, *Glycyrrhiza uralensis*, acute pneumonia, A549 cell, anti-inflammation, pro-inflammatory cytokines

## Abstract

Acute pneumonia is an inflammatory disease caused by several pathogens, with symptoms such as fever and chest pain, to which children are particularly vulnerable. Gancaonin N is a prenylated isoflavone of *Glycyrrhiza uralensis* that has been used in the treatment of various diseases in oriental medicine. There are little data on the anti-inflammatory efficacy of Gancaonin N, and its effects and mechanisms on acute pneumonia are unknown. Therefore, this study was conducted as a preliminary analysis of the anti-inflammatory effect of Gancaonin N in lipopolysaccharide (LPS)-induced RAW264.7 cells, and to identify its preventive effect on the lung inflammatory response and the molecular mechanisms underlying it. In this study, Gancaonin N inhibited the production of NO and PGE2 in LPS-induced RAW264.7 cells and significantly reduced the expression of iNOS and COX-2 proteins at non-cytotoxic concentrations. In addition, in LPS-induced A549 cells, Gancaonin N significantly reduced the expression of COX-2 and pro-inflammatory cytokines, such as TNF-α, IL-1β, and IL-6. Moreover, Gancaonin N reduced MAPK signaling pathway phosphorylation and NF-κB nuclear translocation. Therefore, Gancaonin N relieved the inflammatory response by inactivating the MAPK and NF-κB signaling pathways; thus, it is a potential natural anti-inflammatory agent that can be used in the treatment of acute pneumonia.

## 1. Introduction

Pneumonia, a lower respiratory tract infection disease, is an inflammatory disease caused by pathogens, such as various bacteria, viruses, fungi, and other factors [1,2,3]. Clinical symptoms include fever, cough, phlegm, drowsiness, chest pain, and shortness of breath [4,5]. Despite the advances in antibacterial therapy and improvement in supportive therapy, it is a major infectious disease that is common in all age groups [6]. It is one of the main causes of death in the United States and also has a fatal effect on children in developing countries [7]. In particular, acute pneumonia is a major cause of mortality and disease rates in children under the age of 5, and it has been reported that 1.1 million to 1.4 million children are diagnosed with pneumonia every year [8,9,10]. Children’s acute pneumonia includes aspiration pneumonia and infectious pneumonia, and a recent study showed that the highest incidence and severity occurs at 6 months after birth [11]. Currently, when pneumonia is suspected or confirmed, antibiotics are commonly used, but sometimes symptoms are not relieved, and one of the most important factors is the specific inflammatory response of the host [12,13,14]. Lipopolysaccharide (LPS) present in Gram-negative bacteria is a factor characterized by an extreme inflammatory reaction, leading to inflammatory diseases in various organs [15]. The mechanism by which LPS affects pneumonia is not fully understood, but there is a lot of evidence that it causes inflammation in the lungs, contributing to the development of acute pneumonia [16,17]. In addition, A549 cells, a human type II lung cell line, along with the human bronchial epithelial cell line (BEAS-2B), is the most commonly used cell line for various studies related to respiratory diseases, including pneumonia. Several studies have shown that LPS-treated A549 cells can promote and amplify the expression of inflammatory mediators by activating the nuclear factor kappa light chain enhancer of activated B cells/mitogen-activated protein kinase (NF-κB/MAPK) signaling pathways [18,19,20]. In addition, by LPS induction, the phosphoinositide-3-kinase/protein kinase B/mammalian target of rapamycin (PI3K/Akt/mTOR) signaling pathways are activated to promote cell cycles and inhibit apoptosis, which affects inflammatory reactions [17,21]. Transcription factors such as AP-1 and STAT-3 are also activated to facilitate the production of inflammatory mediator cyclooxygenase-2 (COX-2) [22]. Therefore, modulation of these signaling pathways may help in the treatment of acute pneumonia by suppressing the inflammatory response. Accordingly, natural products that have few side effects and are inexpensive are receiving more attention as potential anti-inflammatory drugs that can improve the inflammatory state of the lungs and many natural products that can alleviate the inflammatory state of the lungs are continuously being discovered [23]. As an example, Labib et al. reported that the essential oil of *Pinus roxburghii* bark containing palmitic acid as the main component could improve the inflammatory state of the lungs [24], and Gao et al. reported that neochlorogenic acid isolated from *Morus alba* L. relieved oxidative stress and inflammatory response in inflamed lungs [18]. In addition, terpenoids, phenylpropanoid glycosides, and polyphenols, which are naturally derived components isolated from plant extracts, showed the ability to alleviate oxidative control damage and inflammatory responses in the lungs [25,26,27].

The roots and rhizomes of *Glycyrrhiza uralensis* are the oldest medicinal herbs in oriental medicine and are mainly prescribed as herbal plants to treat cough, bronchitis, peptic ulcer, and dermatitis [28]. In addition, according to recently published research results, *Glycyrrhiza uralensis* protects the liver from alcoholic fatty liver disease [29], and relieves airway hypersensitivity reactions and oxidative stress, thereby preventing asthma symptoms [30]. In addition, it has been proven to be effective in immunomodulation [31] and has anti-inflammatory [32], anti-obesity [33], anti-viral [34], and antibacterial effects [35]. Therefore, based on the efficacy of *Glycyrrhiza uralensis*, various pharmacological activities have been reported for various compounds (triterpenoids, flavones, isoflavones, among others) derived from *Glycyrrhiza uralensis* [36,37].

Gancaonin N is a prenylated isoflavone-based organic compound extracted from the roots of *Glycyrrhiza uralensis* [38]. In a recent study, Gancaonin N was shown to have antiproliferative activity in human-derived tumor cell lines and an inhibitory effect on the production of nitric oxide (NO) in LPS-induced RAW264.7 cell lines [39]. However, no clear study of the therapeutic mechanism of Gancaonin N in lung inflammation has been revealed. Therefore, in this study, before verifying the effect of Gancaonin N in inflammatory conditions in the lungs, it was confirmed that Gancaonin N inhibits the production of inflammatory mediators, NO and prostglandin E2 (PGE2), and their biosynthetic enzymes, inducible nitric oxide synthase (iNOS) and COX-2, in RAW264.7 cells exposed to LPS. Later, it was confirmed that Gancaonin N has an anti-inflammatory effect in alveolar epithelial A549 cells in the LPS-induced inflammatory state, and further we analyzed its effect on the intracellular inflammatory signal transduction pathway, to suggest the possibility of an acute pneumonia treatment.

## 2. Materials and Methods

### 2.1. Reagents

Gancaonin N was purchased from Chemfaces (Wuhan Chemfaces Biochemical Co., Ltd., Wuhan, China). LPS (*Escherichia coli* 055:B5) was purchased from Sigma-Aldrich (St. Louis, MO, USA). Primary antibodies for COX-2, NF-κB p65, extracellular signal-regulated kinase (ERK), p-ERK, p38, p-p38, and Lamin B1 were purchased from Cell Signaling Technology (Beverly, MA, USA), interleukin (IL)-6, IL-1β, and tumor necrosis factor-α (TNF-α) were purchased from Proteintech (Rosemont, IL, USA), PGE2 was purchased from Bioss (Woburn, MA, USA), and iNOS was purchased from R&D Systems Inc. (Minneapolis, MN, USA). β-Actin and secondary antibodies were purchased from Santa Cruz Biotechnology (Dallas, TX, USA).

### 2.2. Cell Culture

The RAW264.7 cell line was purchased from the Korean Cell Line Bank (Seoul, Korea), while the A549 cell line was purchased from American Type Culture Collection (ATCC, MD, USA). Each cell line was cultured using DMEM containing 10% FBS (Gibco, NY, USA) and 1% antibiotics (Corning Inc., New York, NY, USA) at a temperature of 37°C, 5% CO_2_ environment [40].

### 2.3. Cytotoxicity Assay

To measure the cytotoxicity of Gancaonin N in RAW264.7 and A549 cells, the 3-(4,5-dimethyl-2-thiazolyl)-2,5-diphenyltetrazolium bromide (MTT) assay was performed according to Ko et al.’s paper method described previously [41]. Briefly, cells were seeded in a 96-well plate (1 × 10^4^ cells/well), followed by treatment with Gancaonin N (0–40 μM) for 24 h. The absorbance of formazan produced by the MTT solution was quantified at 540 nm using a 96-well microplate reader (Bio-Rad, Hercules, CA, USA).

### 2.4. NO Assay

NO assay was performed according to our method previously described [42]. Briefly, RAW264.7 cells were seeded in a 6-well plate (6 × 10^5^ cells/well). The cultured medium was exchanged for each Gancaonin N dose 2 h prior to treatment with 1 μg/mL LPS. The cells were then incubated for 24 h. Thereafter, the cultured medium of each well and Griess reagent were mixed and reacted for 10 min, and absorbance was measured at 540 nm.

### 2.5. Immunoblotting

The preparation of whole cell lysates and nuclear fractions and the whole immunoblotting process were performed based on the previously detailed description of Ko et al. [41]. To identify a specific protein band, ImageQuant LAS 500 (GE Healthcare Life Sciences, Sydney, NSW, Australia) was used by treating the EZ-Western Lumi Femto kit (DoGen, Seoul, Korea) and Image J software (NIH, Bethesda, MD, USA) was used to quantify the band [43].

### 2.6. Immunofluorescence Assay

A549 cells were seeded in 4-well culture slides. Cells were pretreated with Gancaonin N at 40 μM 2 h before stimulation with LPS and incubated for 6 h. After that, to confirm that NF-κB p65 was translocated to the nucleus, it was performed according to our previously described method [44,45]. Fluorescence images of each slide were obtained using an EVOSR Cell Imaging system (Thermo Fisher Scientific, Waltham, MA, USA). 

### 2.7. Isolation of the Total RNA and Real-Time PCR

The methods of RNA acquisition, cDNA synthesis, and real-time PCR were previously described in detail [41,42,44]. The relative mRNA expression level of each target gene was normalized with GAPDH [46]. The primer sequences used for real-time PCR analysis are as below. 

TNF-α: (F) 5′-GCAGGTCTACTTTGGGTCATTG-3′ and (R) 5′-GCGTTTGGGAAGGTTGGA-3′.

IL-1β: (F) 5′-TCAGCCAATCTTCATTGCTCAA-3′ and (R) 5′- TGGCGAGCTCAGGTACTTCTG -3′.

IL-6: (F) 5′-AGGGCTCTTCGGCAAATGTA-3′ and (R) 5′- GAAGGAATGCCCATTAACAACAA-3′.

GAPDH (F) 5′-GCCACATCGCTCAGACACC-3′ and (R) 5′-CCCAATACGACCAAATCCGT-3′.

### 2.8. Statistical Analysis 

All data were expressed as mean ± SEM through repeated experiments. An unpaired *t*-test (one-tailed) was used to analyze the statistical significance between each group. 

## 3. Results

### 3.1. Effects of Gancaonin N on RAW264.7 and A549 Cell Viability 

Before analyzing the anti-inflammatory effects of Gancaonin N, the researchers used an MTT assay for evaluating the cytotoxicity of Gancaonin N in RAW264.7 and A549 cells. As shown in Figure 1B,C, when Gancaonin N was used for treatment at a concentration of 5 to 40 μM for 24 h in each cell, it was confirmed that no cytotoxicity was observed. Therefore, we set the concentration of Gancaonin N to up to 40 μM for the subsequent experiments under conditions that did not affect the cells (Figure 1B,C).

### 3.2. Effect of Gancaonin N on Pro-Inflammatory Mediators in RAW264.7 Cells 

Before confirming the acute pneumonia prophylactic effect of Gancaonin N, we evaluated the inhibitory effect of the inflammatory response in LPS-stimulated RAW264.7 cells. NO is an inflammatory mediator and is expressed only when cells are exposed to pro-inflammatory conditions. Therefore, in RAW264.7 cells stimulated with LPS, the amount of NO production was excessively increased compared to that in the untreated group. However, in the group pretreated with Gancaonin N before LPS treatment, NO production significantly decreased, depending on the concentration of Gancaonin N (Figure 2A). The expression of PGE2, a major mediator of chronic inflammatory diseases, was confirmed at the protein level using immunoblotting. As shown in Figure 2B, the level of PGE2 was significantly increased by LPS induction, and it was confirmed that Gancaonin N at a concentration of 10–40 μM significantly suppressed the level of PGE2. Additionally, the effect of Gancaonin N on the expression of pro-inflammatory proteins was analyzed. iNOS is frequently produced in mononuclear phagocytes and among the three types of nitric oxide synthase (NOS)-related enzymes it produces the largest amount of NO, causing severe inflammation. COX-2 is an enzyme that stimulates the synthesis and secretion of prostaglandin E2 (PGE2) by inducing an inflammatory mediator. Therefore, we analyzed the protein expression levels of iNOS and COX-2 (Figure 2C,D). Consequently, Gancaonin N inhibited the expression of iNOS and COX-2 in a treatment concentration-dependent manner in LPS-induced RAW264.7 cells. In particular, the expression level of iNOS was significantly decreased at a Gancaonin N concentration of 5–40 μM, and the expression level of COX-2 was significantly decreased at a concentration of 20–40 μM.

### 3.3. Effects of Gancaonin N on Pro-Inflammatory Cytokine and COX-2 Expression in LPS-Induced A549 Cells

Based on the mechanism of lung inflammation, several studies are being conducted with the purpose of developing therapeutic agents for respiratory diseases, such as pneumonia. Therefore, we confirmed the anti-inflammatory effect of Gancaonin N in LPS induced-A549 cells and evaluated its potential as a therapeutic agent for lung disease prevention. First, we measured the expression levels of pro-inflammatory cytokines using real-time PCR (Figure 3A–C). As a result, it was confirmed that in LPS-induced A549 cells, Gancaonin N significantly reduced the mRNA expression level of each pro-inflammatory cytokine. Moreover, to further confirm the anti-inflammatory effect of Gancaonin N at the protein level, immunoblotting analysis was performed to confirm the effect on the expression of each pro-inflammatory cytokine and COX-2. As shown in Figure 4A–D, it was confirmed that pretreatment with Gancaonin N clearly decreased the protein levels of TNF-α, IL-1β, IL-6, and COX-2, which were increased by LPS induction.

### 3.4. Effect of Gancaonin N on MAPK/NF-κB Signaling Pathway in LPS-Induced A549 Cells

Based on the inhibitory effect of Gancaonin N on the expression of pro-inflammatory cytokines and COX-2, to further analyze the mechanism of anti-inflammatory effects, the protein expression level of the MAPK/NF-κB signaling pathway associated with inflammation was evaluated using immunoblotting. As a result, the phosphorylation of ERK and p38 was increased in A549 cells induced by LPS alone, but the phosphorylation of ERK and p38 was effectively suppressed in the 10–40 μM Gancaonin N pretreatment group (Figure 5A,B). In the nuclear fraction, it was confirmed that the nuclear translocation of NF-κB p65 was inhibited by Gancaonin N in LPS-induced A549 cells (Figure 5C). To further demonstrate the following procedure, we measured the expression of NF-κB p65 in the nucleus using immunofluorescence analysis and we found that 40 μM Gancaonin N significantly inhibited the nuclear translocation of NF-κB p65 via LPS stimulation (Figure 5D). Thus, Gancaonin N regulates the expression of pro-inflammatory cytokines and inflammatory mediators, such as COX-2, through inhibition of the MAPK and NF-κB signaling pathways.

## 4. Discussion

Acute pneumonia, a major respiratory disease, is an inflammatory disease of the lungs caused by various pathogen infections and is a major infectious disease, to which infants are particularly susceptible [47]. Since this disease has a high mortality rate and a high incidence rate, various treatment methods are being developed, along with antibiotic treatment [48]. Among them, compounds derived from natural products that can improve the inflammatory response of the lungs and have relatively few side effects have been attracting much attention for the development of therapeutic agents. Previously, many studies suggested that secondary metabolites such as polyphenols, flavonoids, alkaloids, and terpenoids have the ability to prevent and treat inflammatory responses induced in the lungs [18,24,27,49].

Gancaonin N, a prenylated isoflavone isolated from *Glycyrrhiza uralensis*, has been reported to have anti-inflammatory activity in previous studies [38,39]. However, there are little data on its anti-inflammatory activity and anti-inflammatory mechanisms in pulmonary inflammatory conditions. Therefore, in this study, to evaluate the possibility of preventing and treating pneumonia, the anti-inflammatory effect of Gancaonin N was investigated through experiments using LPS-stimulated RAW264.7 and A549 cells.

Macrophages are the most widely distributed cell type in all tissues in the animal body and are immune cells that play an essential role in innate immune responses, including inflammation in the human body [50]. When macrophages are activated by external stimuli, such as pathogens, they produce various types of inflammatory mediators, such as arachidonic acid metabolites, NO, and pro-inflammatory cytokines [51]. An excessive production of these inflammatory mediators accelerates the development of chronic inflammatory diseases [52]. LPS is a major component of the outer membrane of Gram-negative bacteria and plays a pivotal role in inducing the inflammatory response associated with pneumonia [15,17]. Therefore, many studies have been conducted to analyze the anti-inflammatory effect of drugs by evaluating the inhibitory activity of inflammatory mediators secreted from macrophages activated by LPS stimulation. First, in order to explore the anti-inflammatory effect of Gancaonin N in LPS-stimulated RAW264.7 cells, we examined the inhibitory effect of Gancaonin N on NO and PGE2 production. NO, which is most easily observed in macrophages of inflammatory disease patients, is produced from L-arginine by nitric oxide synthases and contributes to anti-inflammatory activity under normal physiological conditions [53]. However, an excessive production of NO due to a physiological disorder promotes the biosynthesis of inflammatory mediators and intensifies inflammation [53,54]. This leads to serious inflammatory diseases. In addition, PGE2, a key inflammatory mediator, is a major product of COX-2 and actively participates in inflammatory responses, contributing to chronic inflammation and various diseases, including cancer [55,56]. Therefore, chemicals that inhibit NO and PGE2 production are known to have anti-inflammatory effects and may be an alternative strategy for treating inflammatory diseases. In this study, pretreatment with Gancaonin N dose-dependently reduced the NO production, induced via LPS stimulation, which was similar to that reported in a recent study [39]. Therefore, we further investigated whether Gancaonin N can reduce the protein expression levels of iNOS and COX-2. iNOS is a nitric oxide synthase (NOS), an enzyme that produces NO, and unlike endothelial NOS (eNOS) and neuronal NOS (nNOS), it is non-dependent on calcium/calmodulin [57]. iNOS is not normally expressed in cells, but is expressed via transcriptional regulation when it is exposed to external stimulation or pro-inflammatory stimulation, and produces a large amount of NO [58]. Cyclooxygenase (COX) is divided into two isoforms, of which COX-2 is directly involved in the generation of PGE_2_, causing pain and fever, and large amounts are expressed in inflammatory cells via LPS, pro-inflammatory cytokines, growth factors, and tumor promotors [59]. Therefore, the discovery of natural compounds that inhibit the production of NO and PG by inhibiting the expression of iNOS and COX-2 can be an indicator of the development of natural anti-inflammatory drugs with fewer side effects. The results of this study confirmed that the protein levels of iNOS and COX-2 were increased in RAW264.7 cells stimulated with LPS alone, but the protein expression levels of iNOS and COX-2 were suppressed when Gancaonin N was used for pretreatment. The results show that Gancaonin N relieves the inflammatory condition caused by LPS stimulation.

Despite the use of antibiotics to treat pneumonia, sometimes they do not relieve symptoms, and one of the most important factors is the specific inflammatory response of the host [13,14]. Therefore, based on the anti-inflammatory activity of Gancaonin N described above, in order to confirm whether Gancaonin N can help relieve the symptoms of pneumonia by improving the pulmonary inflammatory response, we additionally demonstrated the effect of Gancaonin N on LPS-stimulated A549 alveolar epithelial cells. A549 cells, which have the properties of type 2 alveolar epithelial cells, have been used in many studies to investigate the treatment mechanisms related to lung inflammation due to the limitation of the use of primary cultured human alveolar epithelial cells [60,61,62]. When LPS is used to stimulate A549 cells, the expression of pro-inflammatory cytokines is induced and various inflammatory mediators are produced, making it a widely used pneumonia model because there is evidence that it causes and worsens lung damage [18,19,63]. Therefore, suppressing the production of pro-inflammatory cytokines and COX-2 in LPS-induced A549 cells is a representative treatment method applicable to inflammatory symptoms caused by pneumonia. Inflammatory cytokines, such as TNF-α, IL-1β, and IL-6, are essential cell signaling proteins in the inflammatory response. TNF-α is expressed due to various external stimuli, such as LPS and viruses, and when expressed it causes the influx of inflammatory cells and various biological reactions, such as cell death, proliferation, and migration [64]. IL-1β is a cytokine responsible for initiating and amplifying the inflammatory response and activating NF-κB [65]. IL-6, which plays a role as a major pro-inflammatory mediator for the induction of acute phase reactions, is known to regulate the differentiation and activation of T lymphocytes by inducing the ERK pathway and is considered an important biomarker in respiratory inflammatory diseases [66,67]. As a result of the experiment, we confirmed that the expression of TNF-α, IL-1β IL-6, and COX-2 was upregulated in A549 cells stimulated with LPS alone. This is consistent with previous research findings in A549 cells stimulated with LPS [20]. Furthermore, the pretreatment with Gancaonin N significantly inhibited the expression of TNF-α, IL-1β, IL-6, and COX-2 in LPS-stimulated A549 cells. These results suggest that Gancaonin N inhibits the production of inflammatory mediators produced by LPS stimulation in A549 cells and, thus, has a preventive effect against the inflammatory response.

To characterize these effects, we further investigated whether Gancaonin N could inhibit the activation of MAPK and NF-κB signaling pathways via immunoblotting. MAPK is activated by various inflammatory and stress stimuli and is an intracellular signaling pathway that regulates the immune response by regulating the expression of inflammatory cytokines and other inflammatory mediators in various cells [68]. The MAPK pathway is largely classified into three subtypes: ERK, c-Jun N-terminal kinase (JNK), and p38. Phosphorylation in these pathways can be easily detected in lung diseases related to lung inflammation, such as pneumonia and chronic obstructive pulmonary disease (COPD) [69,70]. Among them, ERK is involved in cell proliferation, growth, differentiation, cell migration, and survival, and is related to pathological conditions, such as cancer and chronic inflammation [71]. The p38 MAPK pathway, which acts as a link for signaling, has a strong association with inflammation and is known to play an essential role in the production of TNF-α, IL-1β, and IL-6 [72,73]. This study showed that Gancaonin N pretreatment inhibited the phosphorylation of ERK and p38, which was increased by LPS stimulation in A549 cells. The NF-κB pathway is an important transcriptional regulator involved in the regulation of the immune system and inflammatory response [74]. NF-κB is present in the cytoplasm in a state bound to IκB in an inactive state; however, when activated by LPS stimulation, NF-κB separates from IκB and moves into the nucleus, inducing the expression of inflammatory cytokines and inducible enzymes, such as COX-2 and iNOS [75]. Therefore, many studies have demonstrated the anti-inflammatory effects of natural drugs on lung inflammation caused by LPS stimulation by inhibiting NF-κB nuclear translocation [18,19,20]. The results of this study show that Gancaonin N significantly inhibits the NF-κB nuclear translocation increased by LPS stimulation in A549 cells. To clarify this, we further confirmed that Gancaonin N clearly inhibited the nuclear translocation of NF-κB p65 in cells induced by LPS stimulation, using immunofluorescence. Therefore, we presume that in LPS-stimulated A549 cells, Gancaonin N exhibits anti-inflammatory effects by inhibiting the activation of the MAPK/NF-κB signaling pathway, which leads to the production of inflammatory mediators.

## 5. Conclusions

We demonstrated the anti-inflammatory activity of Gancaonin N in LPS-stimulated RAW264.7 and A549 cells, providing the potential to prevent acute pneumonia. Gancaonin N was shown to inhibit inflammatory mediators such as NO, PGE2, iNOS, and COX-2 in LPS-stimulated RAW264.7 cells. In addition, in LPS-stimulated A549 cells, Gancaonin N inhibited the expression of TNF-α, IL-1β, IL-6, and COX-2, which is closely related to the inactivation of the MAPK/NF-κB signaling pathway. These findings provide an experimental basis for the anti-inflammatory effect of Gancaonin N; therefore, it has been demonstrated that Gancaonin N may be a natural anti-inflammatory agent that can prevent acute pneumonia. A limitation of this study is that the cytotoxic and anti-inflammatory effects of Gancaonin N were not sufficiently analyzed using lung cell lines other than A549, and anti-inflammatory activity and stability should also be confirmed in an in vivo model in the future.

## Figures and Tables

**Figure 1 pharmaceutics-13-01028-f001:**
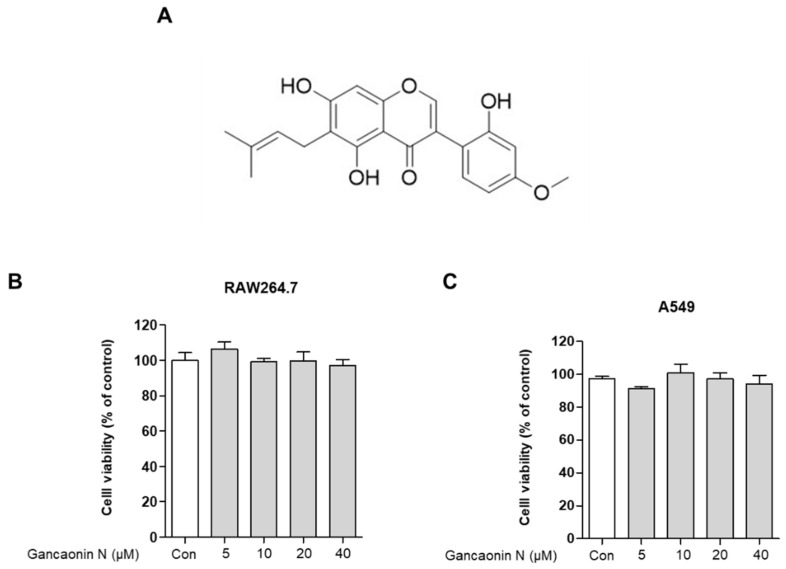
Chemical structure of Gancaonin N (**A**) and its effects on RAW264.7 (**B**) and A549 (**C**) cell viability. Cells were seeded in 96-well plate and treated with Gancaonin N (5–40 μM) for 24 h. The cytotoxicity was confirmed by MTT assay. Values are represented as means ± SEM.

**Figure 2 pharmaceutics-13-01028-f002:**
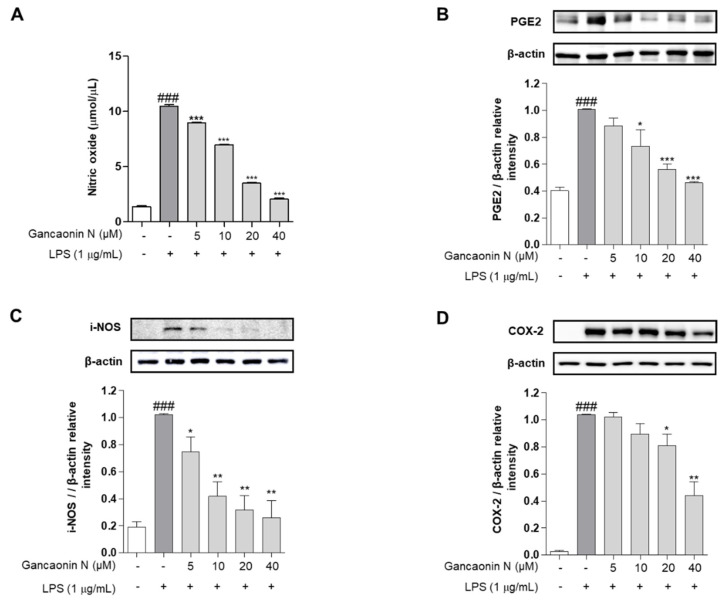
Effects of Gancaonin N on nitric oxide production (**A**) and protein levels of PGE2 (**B**), iNOS (**C**) and COX-2 (**D**) in LPS-induced RAW264.7 cells. Gancaonin N (5–40 μM) was pretreated 2 h prior to LPS-induced inflammation in RAW264.7 cells, incubated for 24 h with LPS (1 μg/mL). The protein levels of PGE2, iNOS and COX-2 were investigated by immunoblotting assay. Ratio of each protein was determined by ImageJ. Values are represented as means ± SEM. ^###^
*p* < 0.001 is significantly different from non-treated group; * *p* < 0.05, ** *p* < 0.01, *** *p* < 0.001 are significantly different from only LPS-treated group.

**Figure 3 pharmaceutics-13-01028-f003:**
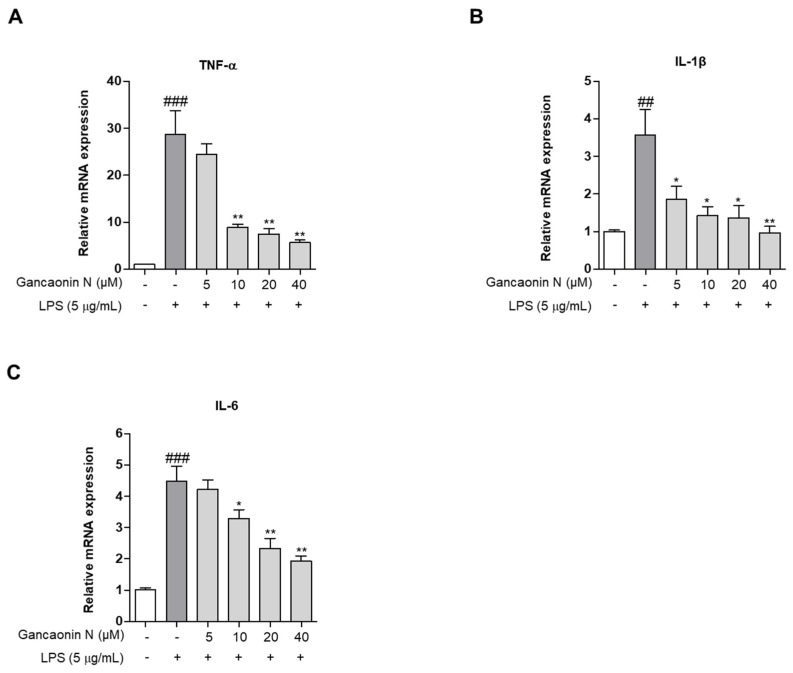
(**A**–**C**) Effects of Gancaonin N on pro-inflammatory cytokine mRNA expression in LPS-induced A549 cells. Gancaonin N (5–40 μM) was pretreated 2 h prior to LPS-induced inflammation in A549 cells, incubated for 24 h with LPS (5 μg/mL). The mRNA expressions of cytokine were investigated by real-time PCR analysis. Values are represented as means ± SEM. ^##^
*p* < 0.01, ^###^
*p* < 0.001 are significantly different from non-treated group; * *p* < 0.05, ** *p* < 0.01 are significantly different from only LPS-treated group.

**Figure 4 pharmaceutics-13-01028-f004:**
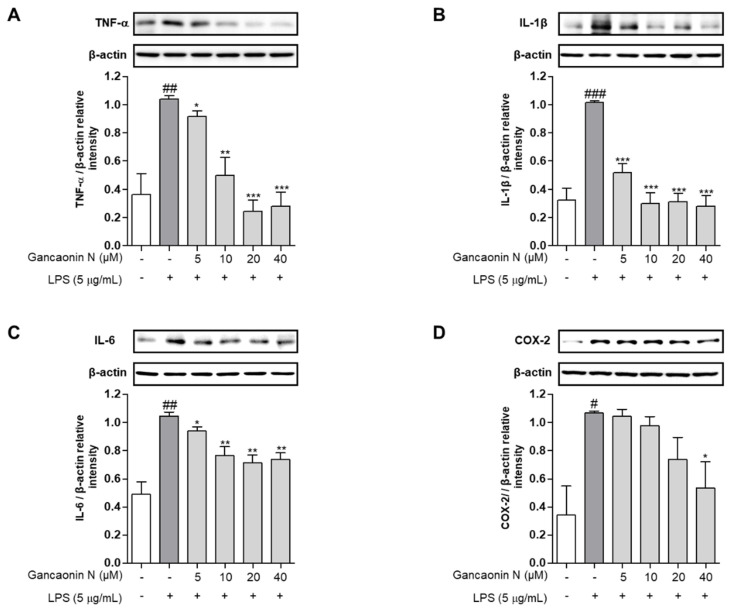
Effects of Gancaonin N on protein levels of pro-inflammatory cytokines (**A**–**C**) and COX-2 (**D**) in LPS-induced A549 cells. Gancaonin N (5–40 μM) was pretreated 2 h prior to LPS-induced inflammation in A549 cells, incubated for 24 h with LPS (5 μg/mL). The protein level of inflammatory biomarkers was investigated by immunoblotting assay. Ratio of each protein was determined by ImageJ. Values are represented as means ± SEM. ^#^
*p* < 0.05, ^##^
*p* < 0.01, ^###^
*p* < 0.001 are significantly different from non-treated group; * *p* < 0.05, ** *p* < 0.01, *** *p* < 0.001 are significantly different from only LPS-treated group.

**Figure 5 pharmaceutics-13-01028-f005:**
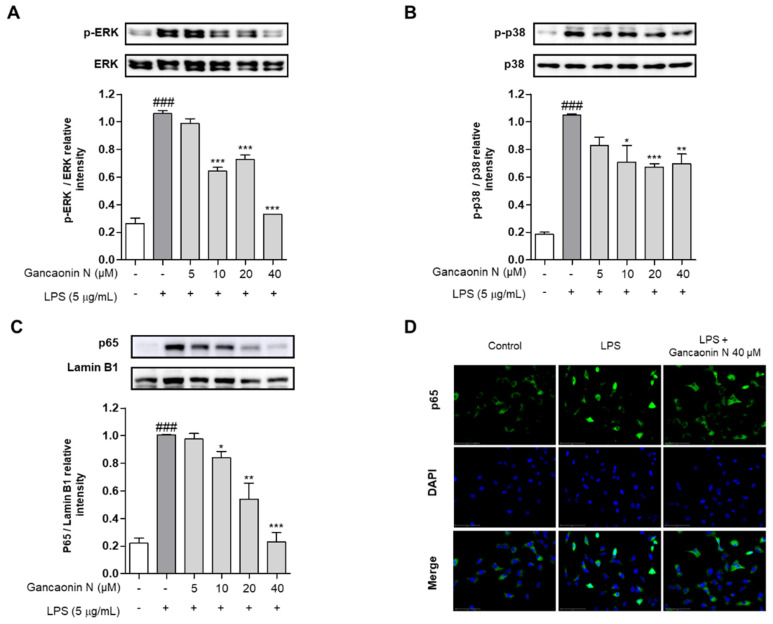
Effect of Gancaonin N on MAPK/NF-κB signaling pathway in LPS-induced A549 cells. Gancaonin N (5–40 μM) was pretreated 2 h prior to LPS-induced inflammation in A549 cells, incubated for 6 h with LPS (5 μg/mL). The protein levels of ERK (**A**), p38 (**B**), and NF-κB p65 (**C**) were investigated by immunoblotting assay. Ratio of each protein was determined by ImageJ. The expression of NF-κB p65 (**D**) in the nucleus was confirmed using immunofluorescence assay (magnification: 400×, scale bar:75 μm). Values are represented as means ± SEM. ^###^
*p* < 0.001 is significantly different from non-treated group; * *p* < 0.05, ** *p* < 0.01, *** *p* < 0.001 are significantly different from only LPS-treated group.

## Data Availability

All data presented this study are available from the corresponding author, upon reasonable request.

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
