# Peer review of "Gancaonin N from Glycyrrhiza uralensis Attenuates the Inflammatory Response by Downregulating the NF-κB/MAPK Pathway on an Acute Pneumonia In Vitro Model"

_pharmaceutics, 2021, doi:10.3390/pharmaceutics13071028_

Round 1

Reviewer 1 Report

The work is well-organized and easy to understand. In the current manuscript, authors thoroughly conducted the experiments to draw the conclusions. However,the limitation of the study in vitro but not in vivo should be discussed.

Author Response

Dear Pharmaceutics Editor,

We would like to thank you for the letter dated 27h May, 2021, and we hereby resubmit a revised copy of the manuscript for your consideration. We would also like to express our thanks to all the reviewers for the positive feedback and helpful comments.

In this revised manuscript, we responded carefully to the comments by reviewer’s point by point as shown in Author’s response. We hope that we addressed well to comments and also wish our revised manuscript would be accepted for publication in Pharmaceutics.

Best regards,

Hyeung-Jin Jang

Reviewer 1

Point 1: The manuscript should be revised for linguistic, grammatical and style errors.

(Response) Thanks for your comments. As pointed out by the reviewer, we revised and improved the manuscript in general.

Point 2: The whole manuscript should be revised for proper use of abbreviations.

(Response) Thank you for your comments. As the reviewer pointed out, we have thoroughly revised it to use proper abbreviations.

Point 3: The interplay between inflammation and other pathways involved should be discussed in more details.

(Response) Thank you for your valuable comments. As pointed out, we added relevant information to the introduction part (page 2, lines 58-63).

Point 4: The manuscript introduction and discussion sections should be enriched with recent literature covering the use of natural products against lung injury.

(Response) Thank you for your valuable comments. As pointed out, we have added some sentences to the introduction and discussion sections based on relatively recent literature dealing with natural product research, including references recommended by reviewers. Thank you for your kind advice to improve the quality of the paper (page 2, lines 64-75; page 8, lines 266-271).

Point 5: The authors are required to write the limitations of the study

(Response) As pointed out by the reviewer, the limitations of the study were written in the conclusion section (page 10, lines 386-389).

Point 6: Why did not used animal model for lung injury?

(Response) We are planning an animal experiment to investigate the efficacy of Gancaonin N as a follow-up experiment.

Thanks for your kind and detailed advice.

Reviewer 2 Report

The manuscript should be revised for linguistic, grammatical and style errors.

The whole manuscript should be revised for proper use of abbreviations.

The interplay between inflammation and other pathways involved should be discussed in more details.

The manuscript introduction and discussion sections should be enriched with recent literature covering the use of natural products against lung injury. 

The following references might be helpful:

DOI: 10.3390/molecules22091384

 doi: 10.3390/biom10030352.

Oxidative Medicine and Cellular Longevity201920198278454

The author are required to write the limitations of the study

Why did not used animal model for lung injury?

Author Response

Dear Pharmaceutics Editor,

We would like to thank you for the letter dated 27h May, 2021, and we hereby resubmit a revised copy of the manuscript for your consideration. We would also like to express our thanks to all the reviewers for the positive feedback and helpful comments.

In this revised manuscript, we responded carefully to the comments by reviewer’s point by point as shown in Author’s response. We hope that we addressed well to comments and also wish our revised manuscript would be accepted for publication in Pharmaceutics.

Best regards,

Hyeung-Jin Jang

Reviewer 2

Point 1:  Since “the discovery of natural compounds that inhibit the production of NO and PG by inhibiting the expression of iNOS and COX-2 can be an indicator of the development of natural anti-inflammatory drugs with fewer side effects," there is a need to provide results of effects of Gancaonin N on PGE2 level consistently in the LPS-induced RAW264.7 cells model. It is especially justified due to the relatively low COX2 response upon the treatment with the test compound - COX-2 was significantly decreased only at higher concentrations of Gancaonin N, i.e., 20 40 μM.

(Response) Thanks for your comments. As pointed out by the reviewer, we additionally evaluated PGE2 levels in the LPS-induced RAW264.7 cells model. As a result, the level of PGE2 was significantly increased by LPS induction, and it was confirmed that Gancaonin N at a concentration of 10 - 40 μM significantly suppressed the level of PGE2. So we added the following figure to Figure 2.

Point 2: The evaluation of the anti-inflammatory effects of Gancaonin N on pro-inflammatory cytokines in LPS- induced A549 cells should consistently include the results of their response at the level of protein expression.

(Response) Thank you for your comments. As the reviewer pointed out, to further verify the anti-inflammatory effect of Gancaonin N at the protein level, immunoblotting analysis was performed to determine the effect on the protein expression of each pro-inflammatory cytokine. As shown in the Figure 4a-d, it was confirmed that pretreatment with gancaonin N clearly decreased the protein levels of TNF-α, IL-1β, and IL-6, which were increased by LPS induction. Therefore, we have added these to the manuscript. Thank you for your careful advice to increase to validity of the data.

Thanks for your kind and detailed advice.

Reviewer 3 Report

The MS by Ko et al. demonstrates the results of a study on the anti-inflammatory activity of Gancaonin N an isoflavone of Glycyrrhiza uralensis in an in vitro acute pneumonia.

The authors, however, should address a few issues, as listed below, before the article could be considered for publication.

  1. Since “the discovery of natural compounds that inhibit the production of NO and PG by inhibiting the expression of iNOS and COX-2 can be an indicator of the development of natural anti-inflammatory drugs with fewer side effects," there is a need to provide results of effects of Gancaonin N on PGE2 level consistently in the LPS-induced RAW264.7 cells model. It is especially justified due to the relatively low COX-2 response upon the treatment with the test compound - COX-2 was significantly decreased only at higher concentrations of Gancaonin N, i.e., 20 40 μM.
  2. The evaluation of the anti-inflammatory effects of Gancaonin N on pro-inflammatory cytokines in LPS- induced A549 cells should consistently include the results of their response at the level of protein expression.

Author Response

Hyeung Jin Jang, Ph.D.

Professor

Department of Biochemistry, College of Korean Medicine

Kyung Hee University, 26, KungHeedae-ro, Dondaemun-gu, Seoul, 02447, Korea

E-mail: hjjang@khu.ac.kr

Jun. 25th, 2021

Dear Pharmaceutics Editor,

We would like to thank you for the letter dated 27h May, 2021, and we hereby resubmit a revised copy of the manuscript for your consideration. We would also like to express our thanks to all the reviewers for the positive feedback and helpful comments.

In this revised manuscript, we responded carefully to the comments by reviewer’s point by point as shown in Author’s response. We hope that we addressed well to comments and also wish our revised manuscript would be accepted for publication in Pharmaceutics.

Best regards,

Hyeung-Jin Jang

Reviewer 3

Point 1: The work is well-organized and easy to understand. In the current manuscript, authors thoroughly conducted the experiments to draw the conclusions. However, the limitation of the study in vitro but not in vivo should be discussed.

(Response) Thank you for your critical comment. As pointed out by the reviewer, the limitations of the study were written in the conclusion section (page 10, lines 386-389).

Thanks for your kind advice.

Round 2

Reviewer 2 Report

As the authors addressed the reviewers comments, I suggest acceptance of the manuscript.